# The Humanised NPY-mRFP RBL Reporter Cell Line Is a Fast and Inexpensive Tool for Detection of Allergen-Specific IgE in Human Sera

**DOI:** 10.3390/diagnostics12092063

**Published:** 2022-08-25

**Authors:** Prema S. Prakash, Nafal J. S. Barwary, Michael H. W. Weber, Daniel Wan, Iván Conejeros, Bernardo Pereira Moreira, Waleed S. Alharbi, Jaap J. van Hellemond, Jude Akinwale, Franco H. Falcone

**Affiliations:** 1Institute for Parasitology, Justus Liebig University Giessen, 35392 Giessen, Germany; 2School of Pharmacy, University of Nottingham, Nottingham NG7 2RD, UK; 3Department of Pharmaceutics, Faculty of Pharmacy, King Abdulaziz University, Jeddah 22254, Saudi Arabia; 4Department of Medical Microbiology and Infectious Diseases, Erasmus MC University Medical Center, 3015 GD Rotterdam, The Netherlands

**Keywords:** NPY-mRFP, RBL, reporter system, IgE, *Echinococcus granulosus*, EgEF-1β/δ

## Abstract

Rat basophilic leukaemia (RBL) cells have been used for decades as a model of high-affinity Immunoglobulin E (IgE) receptor (FcεRI) signalling. Here, we describe the generation and use of huNPY-mRFP, a new humanised fluorescent IgE reporter cell line. Fusion of Neuropeptide Y (NPY) with monomeric red fluorescent protein (mRFP) results in targeting of fluorescence to the granules and its fast release into the supernatant upon IgE-dependent stimulation. Following overnight sensitisation with serum, optimal release of fluorescence upon dose-dependent stimulation with allergen-containing extracts could be measured after 45 min, without cell lysis or addition of any reagents. Five substitutions (D194A, K212A, K216A, K226A, and K230A) were introduced into the FcεRIα cDNA used for transfection, which resulted in the removal of known endoplasmic reticulum retention signals and high surface expression of human FcεRIα* in huNPY-mRFP cells (where * denotes the penta-substituted variant), comparable to the ~500,000 FcεRIα molecules per cell in the RS-ATL8 humanised luciferase reporter, which is a human FcεRIα/FcεRIγ double transfectant. The huNPY-mRFP reporter was used to demonstrate engagement of specific IgE in sera of *Echinococcus granulosus*-infected individuals by *E. granulosus* elongation factor EgEF-1β and, to a lesser extent, by EgEF-1δ, which had been previously described as IgE-immunoreactive EgEF-1β/δ.

## 1. Introduction

Rat basophilic leukemia (RBL) cells were established in 1973 by Eccleston and co-workers [1] and have since been modified for a variety of diagnostic or therapeutic (e.g., drug development) purposes [2], as well as mechanistic studies [3], through stable or transient transfections. Various adaptations have been generated to allow the RBL cells to bind Immunoglobulin E (IgE) from other mammalian species, e.g., human [4], canine [5], or equine [6] IgE.

A new generation of humanised rat basophilic leukaemia (RBL) IgE reporter cell lines was first introduced with the creation of the EXiLE system, a firefly luciferase-based nuclear factor of activated T-cell (NFAT) reporter [7]. These reporters are very sensitive, allowing detection of as little as 15 pg of IgE or 1 fg/mL of egg white protein using appropriate sera of hen egg white allergic donors [7]. Since then, fluorescent variants of the RS-ATL8 reporter have been created [8] and improved [9], which enable their use in solid-phase-based (e.g., allergen array) format [10]. The advantages and disadvantages of RBL-based, humanised IgE reporters have been discussed in detail before [3,11]. Humanised IgE reporter systems have been used to elucidate an intriguing ‘outbreak’ of allergy to wheat hydrolysate containing cosmetic products in Japan [12,13], for the assessment of potential allergenicity of anti-helminthic vaccine candidates [14], to identify exposure to *Ascaris lumbricoides* as a potential cause underlying red meat allergy (Alpha-Gal sensitisation) [15] or to identify potential new allergens in black tiger shrimps [16], with another group suggesting that they may be superior to conventional diagnostic allergy tests in the case of shrimp allergy [17]. Here, we describe the generation and use of a faster, convenient new humanised fluorescent IgE reporter system, which does not require addition of any substrate and is, therefore, suitable for use in, e.g., high-throughput screening assays. The FcεRIα chain, which is needed to enable RBL cells to bind human IgE [18], has been modified by several amino acid substitutions to increase its surface expression. 

The resulting IgE reporter is a humanised IgE reporter with high human FcεRIα surface expression, which can be used for the detection of specific IgE cross-linking by cognate allergens in as little as 45 min, without the need for exogenous substrate addition.

## 2. Materials and Methods

### 2.1. Antibodies and Reagents 

The following antibodies were used in this study: Direct-Blot™ HRP anti-human FcεRIα mouse monoclonal antibody, clone AER-37 (CRA-1) from Biolegend. Anti-human FcεRIα FITC monoclonal antibody, clone AER-37 (CRA-1) from Thermo Fischer Scientific (#11-5899-42). Mouse Monoclonal anti-6x His-tag antibody (clone GT359) from Merck. Goat anti-mouse IgG (Fc specific)–HRP polyclonal antibody (Merck, Cat Nr. A0168). Rat serum was purchased from Biowest (Nuaillé, France) and human IgE from BioPorto (Hellerup, Denmark).

### 2.2. Cell Culture

All cell culture work was carried out using sterile techniques, in class II microbiological safety cabinets as described before [19]. The cells were cultured at 37 °C and 5% CO_2_ in a humidified incubator using Minimum Essential Medium (Merck) supplemented with 10% *v*/*v* heat-inactivated fetal bovine serum, 100 U/mL penicillin, 100 μg/mL streptomycin and 2 mM L-glutamine (RBL Medium). The RBL medium was supplemented with 1 mg/mL G418 (Thermo Fischer Scientific, Waltham, MA, USA) and 0.2 mg/mL of Zeocin (Invitrogen/Thermo Fischer Scientific) to preserve NPY-mRFP and human FcεRIα transgene expression, respectively. The other reporter cells were cultured with addition of 0.2 mg/mL Hygromycin (Merck, Darmstadt, Germany) for RS-ATL8, 20 µg/mL Blasticidin S (Gibco) for NFAT-DsRed, 20 µg/mL Blasticidin S and 0.6 mg/mL Hygromycin for NFAT-DsRed FCER1G.

### 2.3. Construction of the Stably Transfected NPY-mRFP Reporter

The construction of the humanised NPY-mRFP reporter occurred in two steps. A plasmid with the cDNA encoding a Pre-Pro-Neuropeptide Y (NPY)-mRFP fusion protein, driven by the Cytomegalovirus (CMV) promoter, was a kind gift from Ronit Sagi-Eisenberg, Tel Aviv University, Israel. The NPY-mRFP plasmid was stably transfected into RBL-2H3 cells as follows: 2 μg of plasmid dissolved in ≤5 μL was electroporated into 4 × 10^6^ RBL-2H3 cells in 100 μL cell culture medium using a 4 mm cuvette in a Gene Pulser Xcell™ (Bio-Rad, Hercules, CA, USA) set to 250 V/250 μF with an exponential decay protocol. Transfection of pDsRed-Express-N1 (Clontech) was performed in parallel to estimate overall transfection efficiency. After electroporation, 20 μL unaggregated cells was pipetted into each well of clear, flat-bottomed, tissue culture-treated 12-well polystyrene plates (Corning) containing 1 mL RBL medium, which were placed in a cell culture incubator overnight. Next, 1 mg/mL Geneticin (Gibco) was used for stable transfectant selection and to maintain long-term NPY-mRFP expression. After G418 selection, single, stably transfected NPY-mRFP RBL-2H3 cells were sorted into each well of a 96-well plate by FACS using a MoFlo cell sorter (Beckman Coulter, Brea, CA, USA) and a highly fluorescent clone was expanded for further experiments.

In the second step, the stable NPY-mRFP transfectant was transiently transfected with a cDNA encoding a modified FcεRIα chain (FcεRIα*). Modifications were introduced to increase surface expression in the absence of the FcεRIγ chain, as explained in more detail elsewhere [9]. In brief, five amino acids, which have been shown to result in FcεRIα chain retention or retrograde transportation in the absence of FcεRIγ (D192, K212, K216, K226 and K230) [20,21], were substituted with alanine. Additionally, the natural signal peptide was replaced with the cDNA encoding the signal peptide of mouse MHC class I H2-Kb (MVPCTLLLLLAAALAPTQTRAG), as this has also been shown to increase FcεRIα surface expression also in the absence of the FcεRIγ chain [22]. The modified FcεRIα* cDNA was synthesised by GeneArt (Invitrogen/Thermo Fischer Scientific) and subcloned into pcDNA3.1/Zeo plasmid, which contains a Zeocin resistance gene, using *Hin*dIII and *Eco*RV (both from New England Biolabs) as directed by the manufacturer. Successful insertion was verified by DNA sequencing. For transfection, the FcεRIα*-pcDNA3.1/Zeo plasmid was transfected into RBL-2H3 cells as described before [9].

The structural prediction of FcεRIα* was generated with AlphaFold 2 [23] using ColabFold [24].

### 2.4. Flow Cytometry

For staining the surface-expressed human FcεRIα*, cells were seeded in Nunc UpCell Surface cell culture plates (Merck) and incubated overnight, in the appropriately supplemented RBL medium. Cells were then washed once with 1× DPBS w/o Ca^2+^/Mg^2+^ and sensitised with 1 μg/mL human IgE for 16 h at 37 °C. The next day, cells were incubated with 2% rat serum to block the endogenous immunoglobulin receptors for 4 h at 37 °C. For detachment, the cells were kept at room temperature for 30 min, stained using anti-human FcεRIα FITC antibody (1:100 dilution) for 1 h on ice in the dark and washed three times with 1x DPBS. FACS analysis was then performed using a BD Accuri C6 Flow cytometer. Data were analysed with Kaluza Analysis 2.1 software (Beckman Coulter).

### 2.5. Recombinant Expression and Purification of Echinococcus granulosus IgE-Immunoreactive Antigens

The *Echinococcus granulosus* proteins EF1-beta (UniProt U6J0Q2) and EF1-delta (UniProt W6UWE9) were recombinantly expressed in a HEK293-6E suspension cell culture system. The coding sequences for both genes were synthesised by GeneArt (Thermo Fisher Scientific) with restriction sites *Bmt*I/*Bst*EII. In addition to the octa-Histidine tag present in the vector, a c-Myc tag was added in the reverse primer for both genes and cloned into pTT28 vector (National Research Council Canada). The following primers were used: EF-1 beta FW: 5′-GATCGCTAGCATGGTTTTCGGCGATCTCAAG-3′; EF-1 beta RV: 5′-GATCGGTGACCCAGATCCTCTTCTGAGATGAGTTTTTGTTC CAATTTGTTAAAGGCAGCGATATCA-3′; EF-1 delta FW: 5′-GATCGCTAGCATGGAAAGTACGTTGAGGTTTGAT-3′; EF-1 delta RV: 5′-GATCGGTGACCCAGATCCTCTTCTGAGATGAGTTTTTGTTC CAGCTTGTTAAAGGAAGCTACAT-3′.

HEK293-6E cells grown in suspension were transfected with a polyethylenimine:DNA ratio of 3:1, using PEI MAX transfection grade linear 40 kDa polyethyleneimine from Polysciences (Warrington, PA, USA) as described before [15]. The transfected cells were grown in serum-free F17 medium at 37 °C under constant shaking for 5 days. Cells were then centrifuged and the supernatant containing the secreted proteins filtered using 0.45 µm sterile filters. Affinity chromatography purification of the *E. granulosus* recombinant proteins was performed by using a HisTRAP-HP column using an ÄKTA start protein purification system (Cytiva, Marlborough, MA, USA). The purity of the eluted fractions was assessed by stain-free SDS-PAGE gels prepared using 2,2,2 Trichloroethanol (Merck) [25]. The in-frame expression of the proteins was further confirmed by Western blotting using a mouse monoclonal anti-His tag antibody (1:1000) as primary antibody, followed by a polyclonal goat anti-mouse IgG (1:20,000).

### 2.6. Western Blotting

Cells were lysed by heating at 100 °C in Laemmli sample buffer (BioRad) and resolved by 10% or 12% sodium dodecyl sulfate (SDS)-polyacrylamide gel electrophoresis. Proteins were transferred to nitrocellulose membranes using a Transblot Turbo device (BioRad) and blocked with Tris-buffered saline containing 0.1% Tween-20 and 5% skimmed milk powder (blocking buffer). An HRP anti-human FcεRIα was incubated overnight at 4 °C in blocking buffer on a rocking platform. Proteins were detected using Cytiva Amersham™ ECL™ Prime Western Blot detection reagent and ChemiDoc MP imager (BioRad). The signal bands were quantified using BioRad’s Image Lab software against the total protein content.

### 2.7. Extracting Pollen Allergen from Holcus lanatus Grass

*Holcus lanatus* (common velvet) grass was collected in Giessen, Germany and its identity kindly confirmed by B. Honermeier, Department of Agronomy and Crop Physiology, Justus Liebig University of Giessen. The inflorescences were snap frozen in liquid nitrogen, followed by grinding into a fine powder with an autoclaved mortar and pestle. Then, 1.3 g of powder was mixed with 30 mL Ca^2+^/Mg^2+^-free sterile DPBS and stirred in a beaker with a magnetic stirrer for 30 min at room temperature. The supernatant containing the soluble grass extract was collected by centrifugation for 5 min at 1500× *g*, filtered using a 0.22 µm sterile filter, aliquoted, and stored at −20 °C.

### 2.8. Serum Samples

The sera were heat inactivated at 56 °C for 5 min to avoid potential cytotoxicity and added to the cell suspension at 1:50 dilution. This short heat treatment inactivates serum complement without affecting the ability of IgE to bind to the FcεRI receptor [26].

### 2.9. Reporter Assay

Viability was determined by the Trypan blue method using a TC20 automated cell counter (BioRad). First, 100 µL of cells resuspended in culture medium to a density of 1 × 10^6^ viable cells/mL was added to a cell-culture-treated, flat-bottomed 96-well plate and incubated at 37 °C and 5% CO_2_ in a humidified incubator. The cells were sensitised by adding heat-inactivated serum in a 1:50 dilution and incubated for a further 18–20 h. The next day, medium was removed and the cells washed once with DPBS. Using phenol red-free medium (Thermo Scientific Fisher), 65 μL of each of the following conditions was added to the appropriate wells followed by incubation for 45–50 min at 37 °C in a humidified 5% CO_2_ incubator: (1) negative control (cells unstimulated); (2) three positive controls 10 μg/mL Concanavalin A (Merck), 1 μg/mL polyclonal goat anti-human IgE antibody (Merck), *E. granulosus* cyst fluid (10 µg/mL); (3) the recombinant antigens EgEF-1β (0.01–1 µg/mL), EgEF-1δ (0.01–1 µg/mL). After 45–50 min, 50 µL of the supernatant was carefully collected and transferred to 96-well plates (96F nontreated black microwell, NUNC, Thermo Scientific Fisher) for fluorescence measurement using a CLARIOstar Plus multimode microplate reader (BMG LabTech, Ortenberg, Germany).

### 2.10. Statistical Analysis

Fluorescence intensity data are presented as mean ± s.e.m. All data were analysed using GraphPad Prism 8. Statistical significance was defined as a *p*-value ≤ 0.05. Ordinary one-way or two-way ANOVA with multiple comparison tests were used, as detailed in the corresponding figure legends. The obtained p values are denoted using * for *p* ≤ 0.05, ** for *p* ≤ 0.01, *** for *p* ≤ 0.001 and **** for *p* ≤ 0.0001.

## 3. Results

The creation of the humanised NPY-mRFP IgE reporter occurred in two steps. In the first step, RBL-2H3 cells were transfected with the linearized plasmid encoding the pre-pro-NPY-mRFP sequence. The fusion protein is targeted to the granules, resulting in RBL cells containing preformed mRFP in the granules, together with the other mediators [27]. To obtain a stable transfectant, the cells were kept under selective pressure with 1 mg/mL G418 for several weeks. Cells were then separated according to their fluorescence by flow cytometry and a highly fluorescent clone isolated and expanded for further experiments (Figure 1A,B). The resulting cells showed bright expression of red fluorescent protein (Figure 1C), which, as expected, was localised to the granules (Figure 1D).

Next, the NPY-mRFP stable transfectants needed to be transfected with a vector encoding the human FcεRIα chain. The cDNA encoding the FcεRIα chain was modified to remove five known retention signals that result in low surface expression in the absence of FcεRIγ [9]. Furthermore, the natural signal sequence was replaced with the signal peptide of mouse MHC class I H2-Kb shown to increase FcεRIα surface expression [22]. This modified FcεRIα will be designated FcεRIα* henceforth (see Figure 2D).

After transfection, we assessed the cells for the presence of human FcεRIα using Western blotting. Protein extracts of four humanised RBL cell lines RS-ATL8 [7], NFAT-DsRed [8], NFAT-DsRed FCER1G [9], and the humanised NPY-mRFP were separated on 4–20% stain-free TGX SDS-PAGE gels. The proteins were then transferred by electroblotting to a nitrocellulose membrane and incubated with an HRP-conjugated anti-human FcεRIα antibody and visualised using chemiluminescence.

For normalisation, according to total protein content, we performed stain-free SDS-PAGE (Figure 2A), which allows in-gel detection of proteins after crosslinking of 2,2,2-Trichloroethanol (TCE) with tryptophan residues under UV light [28]. Western blotting demonstrated expression of the human FcεRIα chain in all four tested cell lines (Figure 2B), with a major band around the expected MW of 50 kDa and a weaker lower band below 37 kDa, which may represent immature and mature FcεRIα receptors, similarly to what has been described by other groups [29,30], and in line with a range of 45–60 kDa indicated by Ravetch [31]. While the calculated MW for FcεRIα*, including the five substitutions, is 26.77 kDa, human FcεRIα is known to be heavily glycosylated on account of its six N-glycosylation sites, resulting in a ~40–42% increase in its molecular weight [31].

Data were normalised using the total protein content for each lane in the stain-free gel (Figure 2A) and compared to RS-ATL8, which showed the highest expression levels (Figure 2B,C). However, as Western blotting cannot provide any information about whether the detected FcεRIα is present on the cytoplasmic membrane or in intracellular compartments, such as the endoplasmic reticulum, our next step was to use flow cytometry. We assessed surface levels of FcεRIα* in huNPY-mRFP in comparison to FcεRIα expression on RS-ATL8 (FcεRIα/FcεRIγ double transfectant) and the cell lines RBL-2H3 and (non-humanised) NPY-mRFP, from which the humanised reporter cell line was sequentially derived. Figure 2D shows the location of the substitutions; while four substitutions are lysine to alanine swaps located close to the cytosolic C-terminus of the receptor, the fifth substitution with alanine replaces an unusual aspartic acid residue, which maps to the transmembrane helix. Overall, these modifications resulted in very high surface expression of FcεRIα* (Figure 2E), comparable to the high levels found on RS-ATL8 cells, which we previously found to express about 500,000 human FcεRIα receptors per cell [9]. Flow cytometry, however, also revealed the presence of a cell population, which does not appear to express FcεRIα on the surface, as can be expected from a transient transfection.

Our next aim was to determine the best time point for post-stimulation fluorescence measurement. For this aim, huNPY-mRFP cells were sensitised overnight with serum of a grass pollen allergic donor, after which excess serum was removed by washes and the sensitised cells were stimulated with 1 μg/mL polyclonal goat anti-human IgE or 1 μg/mL *H. lanatus* pollen extract. Supernatants were removed at 15, 30, 45, 60, and 120 min after stimulation and measured in a plate spectrofluorometer. Figure 3 shows a statistically significant increase in fluorescence already after 15 min, reaching an optimum after 45 min, then decreasing progressively. Therefore, an incubation time of 45 min was chosen for all further experiments.

Once an optimal incubation time was established, we assessed dose-dependent fluorescence release after incubation of overnight serum-sensitised cells stimulated for 45 min with a polyclonal anti-human IgE antibody or grass pollen extract. As can be seen from Figure 4, both the antibody and the allergen extract induced maximum activation at 1 μg/mL, with statistically significant activation also at 0.1 μg/mL. The activation curve appears bell shaped, with the highest (supra-optimal) concentrations of stimulant leading to lower cellular activation, as expected for IgE-dependent stimulation.

Finally, we wanted to explore the applicability of huNPY-mRFP for the assessment of IgE-mediated allergenicity (defined here as the ability of an antigen to induce activation of the IgE reporter in a specific IgE-dependent way) of individual molecules, rather than complex extracts. For this purpose, we cloned and recombinantly expressed two antigens from the zoonotic parasite *Echinococcus granulosus*, the causative agent of cystic echinococcosis. *E. granulosus* cysts are well known to contain highly allergenic substances, with the potential to induce anaphylaxis upon cyst rupture, for example, during cyst removal surgery [32]. We chose the following two antigens: *E. granulosus* elongation factor 1β (EgEF-1 β; EGR_02060, UniProt U6J0Q2) and EgEF-1δ (EGR_07280, UniProt W6UWE9). *E. granulosus* EgEF-1β/δ has previously been shown to be recognised by IgE in cystic echinococcosis patients [33]. However, since EgEF-1β and EgEF-1δ are encoded by separate genes and have multiple different cell functions, we wanted to use the huNPY-mRFP reporter to assess whether only one or both elongation factors were recognised by specific IgE in cystic echinococcosis patients. HuNPY-mRFP reporter cells were sensitised overnight with sera of cystic echinococcosis patients or healthy control sera and stimulated with three different concentrations of recombinant EgEF-1β and EgEF-1δ, and fluorescence was measured after 45 min. As shown in Figure 5, EgEF-1β activated the huNPY-mRFP reporter in 4/4 tested sera, while EgEF-1δ was less strong in inducing activation, with only 2/4 resulting in activation. No activation was found in any cases with healthy donor sera, except for the ConA-positive control.

These experiments demonstrate how the huNPY-mRFP system developed in this work can be used to assess the ability of suspected IgE-immunoreactive antigens to activate cells by cross-linking FcεRI receptor-bound IgE. Vice versa, by employing known allergens using the same approach with previously uncharacterized patients’ sera, we can systematically and individually determine a patient’s allergenic IgE responsiveness in a diagnostic setting, limited only by the availability of defined IgE-reactive diagnostic proteins. The short incubation time and the lack of need for expensive substrates also makes this reporter system a convenient choice for, e.g., high-throughput screening experiments of inhibitors of this activation pathway.

## 4. Discussion

RBL-based humanised IgE reporter cell lines offer a series of advantages over the older, humanised non-reporter RBL cell lines. Earlier protocols for assessment of IgE-dependent activation were based on the measurement of beta-hexosaminidase activity, which is preformed in granules and released upon degranulation. The enzymatic method dates back to the original description by Leaback and Walker in 1961 [34]. Some groups have attempted to increase sensitivity by using chemicals, such as deuterated water [35] or N-ethylcarboxamidoadenosine [36], for enhancement of degranulation. We did not find any additives to be necessary; however, when measuring fluorescence, it is important to use black plates with low autofluorescence, as we found some products to substantially worsen the signal-to-noise ratio. Key advantages of the huNPY-mRFP reporter system presented here are the lack of need for expensive substrates or a lysis step (as with the RS-ATL8) and the faster incubation time of 45 min. However, there are two possible disadvantages to using the NPY-mRFP reporter. As fluorescence is released during degranulation, this reporter system cannot be used in array format, as it would disconnect the signal from its location on the array. For use on arrays, we developed another humanised reporter system called RBL NFAT-DsRed FCER1G, where fluorescence is cytosolic and can be measured in array format [10]. The second potential issue is inherent to the signal transduction pathway. Using an anti-IgE antibody for stimulation of basophils via the IgE receptor, the optimum concentration for NFAT activation (reflecting cytokine induction) is an order of magnitude lower than for histamine release and degranulation [37]. Thus, it is possible that degranulation-based humanised RBL IgE reporters are intrinsically less sensitive than the corresponding NFAT reporters.

We previously showed that different humanised IgE reporters, all derived from RBL-2H3 cells, have very different levels of IgE receptors on the cell surface [9]. The highest expression was shown by RS-ATL8 cells, a luciferase reporter obtained from SX-38 RBL cells; both are double FcεRIα/γ chain transfectants and showed approximately 0.5 × 10^6^ FcεRIα surface molecules per cell. FcεRIα surface expression levels in NFAT-DsRed reporter cells were 30-fold lower [9]. This may be explained by the absence of a human FcεRIγ chain in NFAT-DsRed cells, leading to lower FcεRIα surface expression, due to the presence of ER-retention signals in the IgE binding chain that are masked when the two different chains bind to each other [9,20,21,38]. The quantitative Western blotting results shown in Figure 2C seem to reflect a three-fold difference between RS-ATL8 cells and NFAT-DsRed cells and an approximately two-fold difference between RS-ATL8 and huNPY-mRFP cells. This suggests that the overall amount of FcεRIα in the cell is less variable than on the surface. The fact that a substantial fraction of the huNPY-mRFP cells shows similar surface FcεRIα levels (in fact, slightly higher) in comparison with the RS-ATL8 cells (Figure 2E), clearly demonstrates that removal of the five known retention signals (Figure 2D) enables the FcεRIα receptor chain to reach the cell surface, even in the absence of human FcεRIγ chain transfection. Interestingly, we obtained only a modest increase in surface expression by stably transfecting the NFAT-DsRed reporter cell line with the human FcεRIγ chain [9]. This suggests that the association of the two different chains in the ER is only partially able to rescue surface expression of the IgE binding FcεRIα chain.

While selection for stable transfectants with the modified human FcεRIα*chain is still ongoing and will include two subsequent rounds of clonal selection, the NPY-mRFP reporter gene transfection is stable. We did not see any changes in expression of fluorescence in the granules after several months of culture in the presence of selective pressure. Cells recover well upon defrosting after prolonged storage in liquid nitrogen.

Our experiments confirmed the ability of EgEF-1β to cause reporter activation with 4/4 sera of infected individuals, while EgEF-1δ only activated the huNPY-mRFP reporter sensitised with 2/4 tested sera. No activation was seen with any of the tested healthy donor sera. Optimal concentration differed between individuals, a well-known feature also found in activation of peripheral blood basophils from different individuals [39]. This also illustrates the necessity to use a range of concentrations for each tested IgE-immunoreactive antigen. Finally, it is also apparent from our data that the signal levels obtained with EgEF-1β and -1δ are weaker than those obtained with the cyst extract. This suggests that the cyst extract contains more powerful, yet unknown IgE-immunoreactive antigens and/or that the signal strength is the result of the additive effects of multiple IgE-immunoreactive antigens. However, any gain in sensitivity obtained by using cyst extract for diagnosis would need to be weighed against the increased potential for cross-reactivities with related parasite species, translating into lower specificity. Thus, we believe that the best compromise may be achieved by carefully choosing a small set of diagnostic antigens with the best signal strength and the lowest cross-reactivity with other parasitic species.

## 5. Conclusions

Taken together, we generated a new, fast, humanised IgE reporter cell line, which is characterised by high-FcεRIα surface expression, probably due to the five amino acid substitutions introduced mainly in the intracellular region of the protein. The huNPY-mRFP reporter only requires a short incubation time of 45 min (which compares favourably with the >16 h needed for the NFAT-DsRed reporter) and does not require the addition of any substrates. This makes it a suitable candidate for use in, e.g., high-throughput screening for degranulation inhibitors, but also for IgE-based diagnostic purposes. It will be interesting to validate the new reporter with a larger number of sera and compare its suitability for the diagnosis of allergic sensitisation or parasitic infection, comparing it to, e.g., basophil activation tests or other tests used in clinical practice.

## Figures and Tables

**Figure 1 diagnostics-12-02063-f001:**
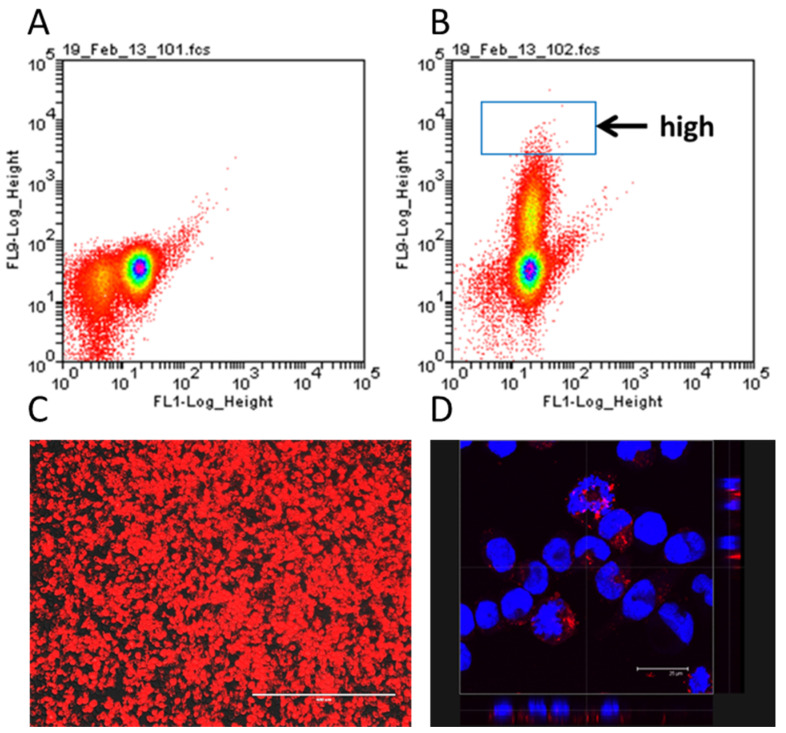
FACS assessment of NPY-mRFP RBL-2H3. (**A**) Dot-plot of untransfected RBL-2H3 cells. (**B**) Dot-plot of stably transfected NPY-mRFP RBL cells. Highly fluorescent cells were gated for cloning (see blue square). (**C**) Fluorescence microscopy showing unstimulated NPY-mRFP cells with preformed fluorescence in the granules (100× magnification, bar size 400 μm). (**D**) Confocal microscopy of NPY-mRFP cells obtained from the clonal, highly fluorescent NPY-mRFP cells after sorting. Red fluorescence is from the NPYmRFP fusion protein, blue is nuclear DAPI stain. The bottom line and left side show the Z-stack (magnification 630×, bar size 20 μm).

**Figure 2 diagnostics-12-02063-f002:**
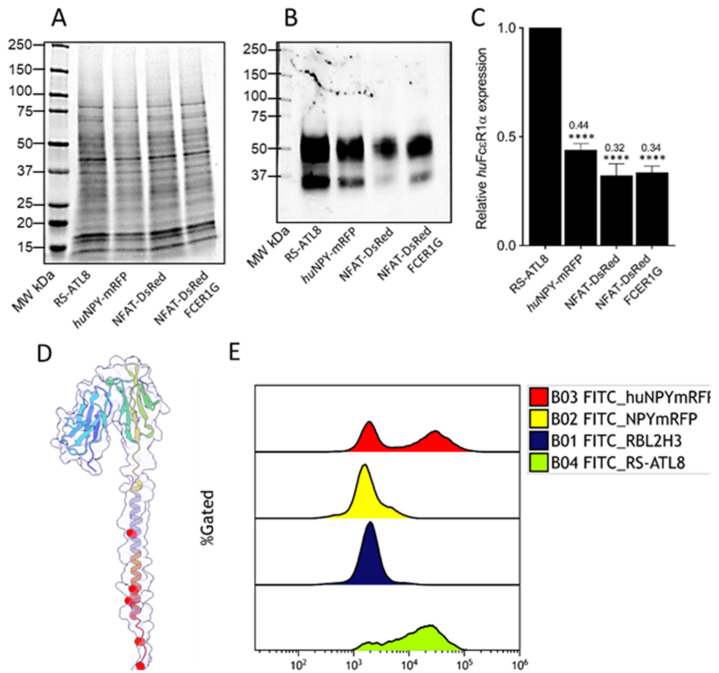
Comparison of human FcεRIα transgene expression in humanised NPY-mRFP and RS-ATL8 cell lines. Cell extracts of four different humanised RBL reporters were separated in 4–20% gradients stain-free SDS-PAGE gels (**A**). The same extracts were transferred to a nitrocellulose membrane and probed with an HRP-conjugated human FcεRIα-specific antibody (**B**). Intensity of bands was normalised to total protein content using the corresponding lanes in the stain-free gel and expressed as relative expression units compared to the RS-ATL8 cell line with the highest expression. Statistical analysis was performed using ordinary one-way ANOVA followed by Dunnett comparison test, with **** indicating *p* ≤ 0.0001. (**C**). AlphaFold2-generated structural model of the human FcεRIα* receptor (**D**). The transmembrane segment of the alpha-helix is coloured in grey. The five positions (D194A, K212A, K216A, K226A, K230A) which were mutagenised to remove known retention/retrograde transportation signals, known to reduce surface expression of the FcεRIα subunit [9], are highlighted using red spheres. (**E**) Histograms comparing human FcεRIα levels on RBL-2H3 (dark blue), non-transfected NPY-mRFP (yellow), huFcεRIα-transfected NPY-mRFP cells (red) and the RS-ATL8 humanised IgE reporter (light green) stained with anti-human FcεRIα FITC.

**Figure 3 diagnostics-12-02063-f003:**
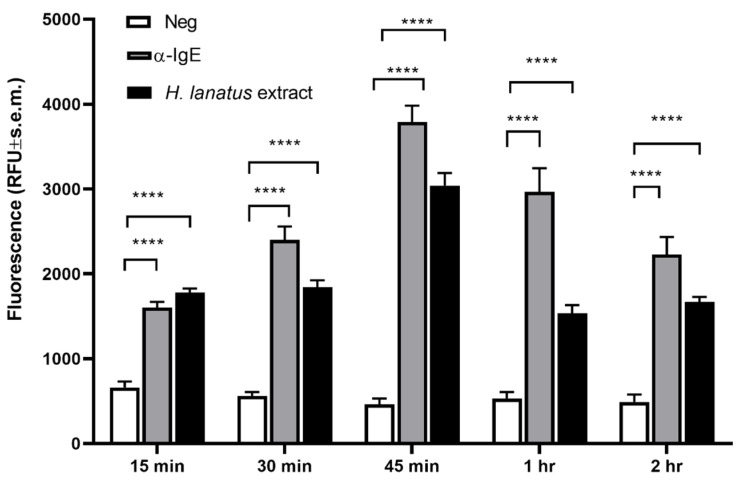
Time course of huNPY-mRFP sensitised with serum of a grass pollen allergic donor and stimulated with a polyclonal antibody against human IgE or a *H. lanatus* pollen extract, both at 1 μg/mL. Supernatants were collected at the indicated time points and the released fluorescence was measured. Shown is the mean fluorescence ± s.e.m. from three independent experiments, each performed in triplicate. Statistical analysis was performed using two-way ANOVA followed by Tukey multiple comparison test. **** indicates *p* ≤ 0.0001.

**Figure 4 diagnostics-12-02063-f004:**
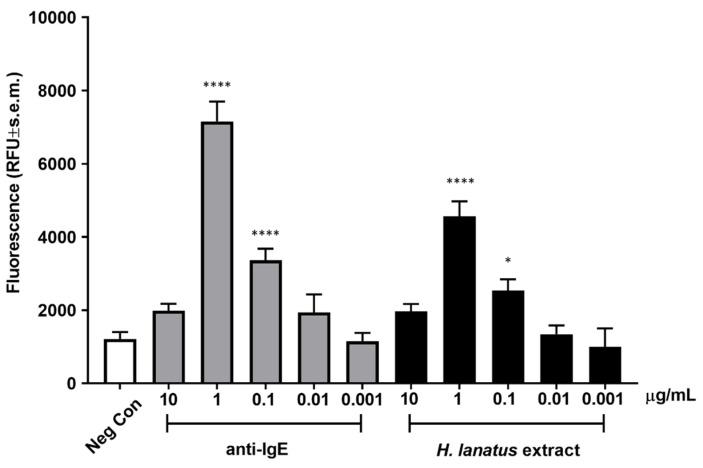
Dose-dependent activation of huNPY-mRFP cells upon 45 min exposure to polyclonal anti-human IgE antibody or *H. lanatus* pollen extract after overnight sensitisation with grass pollen allergic donor serum diluted 1:50. The mean fluorescence ± s.e.m. from three independent experiments, each performed in triplicate is shown. Statistical analysis was performed using ordinary one-way ANOVA followed by Dunnett multiple comparison test. The obtained p values are shown using * for *p* ≤ 0.05 and **** for *p* ≤ 0.0001.

**Figure 5 diagnostics-12-02063-f005:**
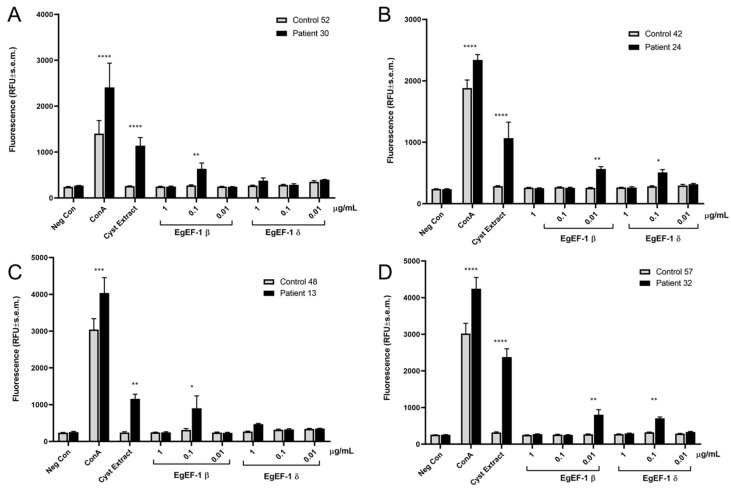
Results of huNPY-mRFP activation after sensitisation with 4 different cystic echinococcosis sera (Nr. 30, 24, 13, and 32 in (**A**–**D**), respectively) and 4 different healthy individuals’ sera (Nr. 52, 42, 48, and 57 in (**A**–**D**), respectively). Cells were sensitised overnight with 1:50 dilutions of sera and left unstimulated (Neg Con), stimulated with the positive control ConA (1 μg/mL), *E. granulosus* cyst extract (1 μg/mL), or 3 different concentrations of recombinant EgEF-1β or EgEF-1δ. Statistical analysis was performed by two-way ANOVA followed by Bonferroni comparison test comparing control with patient serum for each condition. The obtained p values are denoted using * for *p* ≤ 0.05, ** for *p* ≤ 0.01, *** for *p* ≤ 0.001 and **** for *p* ≤ 0.0001.

## Data Availability

All data generated and analysed during this study are included in this published article. Raw data supporting the findings of this study are available from the corresponding author on request.

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
