# Peer review of "The Humanised NPY-mRFP RBL Reporter Cell Line Is a Fast and Inexpensive Tool for Detection of Allergen-Specific IgE in Human Sera"

_diagnostics, 2022, doi:10.3390/diagnostics12092063_

Round 1
Reviewer 1 Report
In this study, the authors describe a new humanized NPY-mRFP RBL-derived reporter cell line for fluorescent detection of allergen-specific IgE in human sera. The system is rapid and does not require the addition of any substrates.
Major points:
Figure 2 shows the results of huFceRI transient transfectants (as mentioned in lines 254/255) with many huFceRI negative cells. However, the authors should also show or at least discuss the results from stable cell lines and the stability of the transfected genes (in weeks or months in culture after transfection in the presence of selective pressure). Furthermore, the authors should show or discuss how the expression of the human FceRIa subunit and huNPY-mRFP changes after freezing and thawing the cells.
Minor points:
Please remove discrepancies:
Anti x anti: compare, e. g., writing at line 126 x 153
NPYmRFP x NPY-mRFP: compare, e. g., writing at line 212 x 216
min x min. x minutes: compare, e. g., writing at line 296 x290 x 279
On lines 220/221, changes in the signal peptide should also be mentioned.
References should be checked, e. g., ref 235 on line 504, ref. 5, 8, 18, 20, 36, ref 38, and many others.
Author Response
We thank the reviewer for the important comments, which have been addressed as follows:
Major point:
While the NPY-mRFP reporter has been stably transfected, we are still in the process of selecting a stable transfectant with the modified human FceRI-alpha* chain. This will include two subsequent rounds of cloning followed by selection of the best performing clones. Based on our experience with the previously generated NFAT-DsRed reporter, this may take as long as one year to complete.
We have added the section (Lines 401-405 in the revised manuscript) "While selection for stable transfectants with the modified human FcεRIα* chain is still ongoing, and will include two subsequent rounds of clonal selection, the NPY-mRFP reporter gene transfection is stable. We have not seen any changes in expression of fluorescence in the granules after several months of culture in the presence of selective pressure. Cells recover well upon defrosting after prolonged storage in liquid nitrogen."
Minor points:
- Anti- is now spelled as anti- throughout the revised m/s
- NPYmRFP now spelled NPY-mRFP
- min. or minutes are now all written as min except in the abstract, where it is spelled out as minutes
- Changes to the signal peptide. We have added the section (Lines 229-231 in Revised m/s): "Furthermore, the natural signal sequence has been replaced with the signal peptide of mouse MHC class I H2-Kb shown to increase FcεRIα surface expression [22]. " This is described is some more detail in the M&M, but we fully agree that it should be mentioned here as well, as this may play a role in improving transgenic receptor surface expression.
- Thank you for pointing this out. We failed to check the reference list after updating the bibliography (which reintroduces some undesired statements such as available at... and accessd on...). The reference list has now been checked again and amended as requested
Reviewer 2 Report
The article is deal with the humanized NPY-mRFP RBL reporter cell line as a fast and inexpensive tool for the detection of allergen-specific IgE in human sera. The topic discussed is very important for the diagnostic of allergic diseases.
I would like to make a few comments:
1) In the title the word humanised should be replaced on: humanized
2) line 110: “The thus modified FcεRIα* cDNA was synthesized…”
Comment:
The word “thus” better to delete.
3) line 298: “Shown is the mean fluorescence ± s.e.m. from three independent experiments, each performed in triplicates”
Comment:
Perhaps, it is better to wright:
The mean fluorescence ± s.e.m. from three independent experiments, each performed in triplicates is shown.
4) The item “Data Availability Statement” should be deleted.
Author Response
We thank the reviewer for the comments, which have all been addressed as suggested. The Data availability statement was replaced with: "All data generated and analyzed during this study are included in this published article. Raw data supporting the findings of this study are available from the corresponding author on request."